# The Change of Lumbar Spinal Stenosis Symptoms over a Six-Year Period in Community-Dwelling People

**DOI:** 10.3390/medicina57101116

**Published:** 2021-10-16

**Authors:** Koji Otani, Shin-ichi Kikuchi, Shoji Yabuki, Takuya Nikaido, Kazuyuki Watanabe, Kinshi Kato, Hiroshi Kobayashi, Shin-ichi Konno

**Affiliations:** Department of Orthopaedic Surgery, Fukushima Medical University School of Medicine, 1 Hikari-gaoka, Fukushima City 960-1295, Japan; shinichi.kikuchi@mt.strins.or.jp (S.-i.K.); yabuki@fmu.ac.jp (S.Y.); tnikaido@fmu.ac.jp (T.N.); kazu-w@fmu.ac.jp (K.W.); kinshi@fmu.ac.jp (K.K.); hiroshik@fmu.ac.jp (H.K.); skonno@fmu.ac.jp (S.-i.K.)

**Keywords:** lumbar spinal stenosis, epidemiology, natural history, quality of life, predictive factors, comorbidities

## Abstract

*Background and Objectives*: The high prevalence of lumbar spinal stenosis (LSS) and its negative impact on quality of life in the elderly is well known. However, the longitudinal time course of LSS symptoms remains unclear. The purpose of this study was to clarify the longitudinal time course and associated factors of LSS symptoms over a period of six years in a community. *Materials and Methods*: This study was conducted with data prospectively collected in 2004 and 2010 under a retrospective design. In 2004, 1578 subjects (age range: 40 to 79 years) were interviewed on LSS symptoms using a specially designed and validated questionnaire. In 2010, a follow-up study was performed by mail, to which 789 subjects of the 2004 study population responded. Considering that the presence of osteoarthritis (OA) of the knee or hip may influence the participants’ answers in the questionnaire, analysis was performed in all 789 subjects with and 513 subjects without either knee or hip OA. Changes in LSS symptoms between the initial and the 6-year survey were investigated. Multiple logistic regression analysis was used for detecting the risk factors for LSS symptom presence at the six-year follow-up. *Results*: 1. At the six-year follow-up, more than half of the subjects who showed LSS symptoms at the initial analysis became LSS-negative, and 12–15% of those who were LSS-negative became LSS-positive. 2. From the multiple logistic regression analysis, a lower Roland-Morris Disability Questionnaire (RDQ) score and a positive LSS symptom at the initial analysis were detected as predictive factors of the presence of LSS symptoms at the six-year follow-up in the total number of subjects, as well as just in those who did not have either knee or hip OA. *Conclusions*: More than half of the subjects who were LSS-positive at their initial assessment still experienced improvement in their symptoms even after 6 years. This means that both LSS symptoms and their time course vary from person to person. Predictive factors for the presence of LSS symptoms during the six-year follow-up period were RDQ score and positive LSS symptoms.

## 1. Introduction

The high prevalence of lumbar spinal stenosis (LSS) and its negative impact on quality of life in the elderly is well known [1]. Therefore, LSS is quite an important clinical syndrome to protect and maintain the elderly’s health. There are several reports regarding the time course of LSS in clinics or hospitals [2,3,4,5,6,7,8,9,10]; however, because of selection bias and relatively short follow-up periods, the time course of LSS is still not yet fully understood. Since physicians often see patients with LSS, it may be important for these physicians to be aware of changes in LSS symptoms over years.

Previously, we reported on a one-year follow-up of LSS symptoms based on self-reported information alone in a regional community [11]. This report revealed that: (1) more than half of the subjects who were initially clinically diagnosed as having LSS symptoms were LSS symptom negative at the one-year follow-up; (2) 10% of those who were clinically diagnosed as being LSS symptom negative at the initial analysis were LSS symptom positive at the follow-up; (3) the predictive factors for LSS symptom presence at the end of one-year follow-up period could not be clearly detected other than the presence of LSS symptoms and lower Roland–Morris Disability Questionnaire (RDQ) score at the initial analysis [11]. The purpose of the current study was to clarify the six-year time course of LSS symptoms based on self-reported information alone and QoL, as well as the possible predictive factors for LSS symptom presence at the end of six-year follow-up period in a community setting.

## 2. Materials and Methods

This study was approved by the ethical committee of Fukushima Medical University (No. 295, 673). 

### 2.1. Study Design

This longitudinal six-year follow-up study was conducted with data prospectively collected in 2004 and 2010 under retrospective design.

### 2.2. Participants

In 2004, 1578 subjects (562 males and 1016 females) aged 40 to 79 years provided written informed consent to be interviewed and participated in our study. Details were previously reported [1,11]. In brief, these subjects, including those with a history of cerebral infarction or bleeding without operation, were considered eligible for this study. On the other hand, subjects were excluded if they were unable to walk independently, fill out the questionnaires by themselves, had ever undergone brain or spinal surgery, or had experienced a fracture of the lower extremities in the year previous to 2004. In 2010, a six-year follow-up survey was conducted by mail. Of the original 1578 subjects, 789 (262 males and 527 females) returned the questionnaires and could be followed-up with (50% follow-up rate) (Figure 1).

### 2.3. Assessment

LSS symptom presence was determined by a validated LSS diagnostic support tool, which was a self-administered, self-reported history questionnaire (LSS-SSHQ) as described previously [1,11,12,13,14] (See Appendix B). The LSS-SSHQ is reliable for the diagnosis of symptomatic LSS based on self-reported patient information alone [13]. In the current study, those clinically diagnosed as having LSS symptoms according to their LSS-SSHQ answers were categorized as LSS-positive, and those who were not diagnosed as having LSS symptoms were categorized as LSS-negative. In addition, those with any surgical history for LSS within the six-year period were categorized as LSS-positive, regardless of the assessment of the LSS-SSHQ. At the same time, the 36-Item Short Form Health Survey (SF-36) [15,16,17] for the measurement of general health-related QoL (HR-QoL) and the RDQ [18,19,20] for the measurement of LBP-related QoL were also used for assessment. Norm-based scores were used to compare HR-QoLs between the initial analysis and the six-year follow-up [21]. Additionally, because age significantly influences on the incidence and severity of LBP, norm-based RDQ scores were also used to assess the six-year change [22]. Both norm-based SF-36 and RDQ scores are available for people aged 20 to 79 years old. Fifty points is the national average for each item, while more than fifty points indicates a higher-than-average QoL, and less than fifty points indicates a lower-than-average QoL. 

To accurately evaluate the possible predictive factors for LSS symptom presence at the end of six-year period, the factors used at the one-year follow-up in our previous study were used [11]. These factors were treatment for hypertension, cardiovascular disease, cerebrovascular disease, respiratory disease, or diabetes mellitus as comorbidities, pack-year smoking history, depressive symptoms using MH scores from the SF-36 [21], and the presence of osteoarthritis (OA) of the knee and hip [21,22]. Detailed assessment methods were as described in previous articles [1,11]. In addition, considering that the presence of OA of the knee or hip may influence the participants’ answers in the LSS-SSHQ, the present study conducted various analyses in two ways: all participants (*n* = 789) and those who did not have knee and hip OA (*n* = 513).

### 2.4. Statistical Analysis

All statistical analyses were conducted in the same way as the one-year follow-up [11]. Differences or changes in RDQ scores and the eight SF-36 domains between the LSS-positive group and the LSS-negative group at the beginning and end of the six-year period were assessed using the Wilcoxon signed-ranks test. Other statistical analyses regarding RDQ scores and the eight SF-36 domains were conducted by the Mann–Whitney’s U test. Related factors for LSS symptom presence at the initial analysis and predictive factors for LSS symptom presence at the end of the six-year period were examined using a multivariate logistic regression analysis. The risk of LSS symptom development at the six-year follow-up in the LSS-negative group at the initial analysis was assessed using the chi-square test by comparing those who answered “yes” to one or more of questions 1–6 (Q1–6) in the LSS-SSHQ and those who gave no positive answers in 2004; Q1–6 were used to evaluate leg symptoms such as leg pain and numbness. 

In addition, the operation rate for LSS in each of the LSS-positive and LSS- negative groups at the initial analysis were also assessed using the chi-square test. Predictive factors for surgery due to LSS at the end of the six-year period were also examined using a multiple logistic regression analysis. 

All statistical analyses were performed using the StatView software package (version 5.0, SAS Institute Inc., Cary, NC, USA). A *p*-value of less than 0.05 was considered statistically significant.

## 3. Results

### 3.1. Participants

The characteristics were proportionally similar between the 1578 subjects at the initial analysis and the 789 subjects at the six-year follow-up and between the 1079 subjects without either knee or hip OA at the initial analysis and the 513 subjects without either knee or hip OA at the six-year follow-up. Although the proportion of subjects in their 60s in 2010 was higher than in 2004, there were no statistical differences in the proportion of each age group and gender between the initial and six-year follow-up among all subjects and those without either knee or hip OA. With the exception of knee or hip OA, the proportion of older subjects was decreasing (Table 1). The proportions of all demographic characteristics in 2004 and 2010 of all subjects and those without either knee or hip OA had no statistical differences (Table 2).

### 3.2. Time Course of LSS-Positive and LSS-Negative Groups over a Six-Year Follow-Up

Regarding all 789 subjects, 161 (20%) were categorized as LSS-positive and 628 (80%) were categorized as LSS-negative at the initial analysis. At the six-year follow-up, 167 (21%) and 622 (79%) subjects were categorized as LSS-positive and LSS-negative, respectively. The time course of the LSS symptoms were found to be similar between all subjects and those without knee or hip OA at the 6-year follow-up.

Among all 789 subjects, 23 subjects underwent operations for LSS during the six-year period (2.9%). Eleven of these were in the initial LSS-positive group (6.8%), and the remaining twelve were in the initial LSS-negative group (1.9%). Thus, there was a statistical difference in the operation rate between the two groups (*p* = 0.0009) (Figure 2 and Appendix A). 

The time-course pattern varied with age. Among the subjects in their 60s and 70s who were LSS-positive at the initial analysis, approximately 60% of all subjects and 65–70% of those without either knee or hip OA became LSS-negative at the six-year follow-up. This decrease is significantly larger than that of the subjects in their 50s. On the other hand, LSS symptoms most frequently developed at the six-year follow-up in the subjects in their 70s among all age groups who were LSS-negative at the initial analysis (22.7% of all subjects and 17.5% of those without either knee or hip OA) (Table 3). These results were similar between all subjects and those without either knee or hip OA in 2004 and were also similar to those at the previous one-year follow-up (Appendix A [11]. No statistical relationship was observed between gender and time-course in the LSS-positive and LSS-negative groups at the six-year follow-up of both groups (Table 1, Table 3 and Appendix A).

### 3.3. Changes in LBP-Related QoL of the Subjects with or without LSS Symptoms

The norm-based RDQ score in the LSS-positive group was much lower than that in the LSS-negative group at the initial analysis. In all subjects, the score of the LSS-positive group was 48.3 ± 9.3 and that of the LSS-negative group was 55.7 ± 6.5 (*p* < 0.0001). However, in the subjects who switched from being LSS-positive at the initial analysis to being LSS-negative at the six-year follow-up, their norm-based RDQ score improved. To the contrary, in the subjects who moved from being LSS-negative at the initial analysis to being LSS-positive at the six-year follow-up, their norm-based RDQ score worsened. In addition, the norm-based RDQ score of the subjects who showed the same LSS symptoms at the six-year follow-up did not show any change or showed rather improved scores. These findings were almost the same between all subjects and those without either knee or hip OA and were similar to those at the previous one-year follow-up [11].

From a time-course point of view, subjects who were LSS-positive at the initial analysis but negative at the six-year follow-up had a higher norm-based RDQ score at the initial analysis than those who were LSS-positive at both the initial analysis and the six-year follow-up (Table 4). Moreover, subjects who were LSS-positive at the six-year follow-up but negative at the initial analysis showed a lower norm-based RDQ score at the initial analysis than those who were LSS-negative at both the initial analysis and the six-year follow-up. However, statistical differences were only observed in all subjects (Table 4 and Appendix A).

### 3.4. Changes in Norm-Based HR-QoL with the Time Course of the Subjects with or without LSS Symptoms

In the subjects who switched from LSS-positive to negative in 2010, seven of eight domains of the SF-36 in all subjects decreased, contrary to LBP-related QoL. Indeed, three of the eight domains (PF; physical functioning, VT; vitality, SF; social functioning) statistically worsened (Table 5). Among the subjects who were LSS-positive in both the initial analysis and the six-year follow-up, all eight domains decreased, while only PF, RP (role-physical), and RE (role-emotional) statistically worsened (Table 5). These findings suggest that once LSS symptoms appear, HR-QoL tends to worsen regardless of whether the LSS symptoms are improved or persisted. These results were almost the same as the subjects without either knee or hip OA and the previous one-year follow-up (Appendix A) [11]. 

Regarding the subjects who were LSS-negative at the initial analysis but positive at the six-year follow-up, all eight domains in all subjects decreased while PF, RP, BP (bodily pain), VT, and SF in all subjects statistically worsened (Table 5). On the other hand, the subjects who were LSS-negative both at the initial analysis and the six-year follow-up, seemed to maintain HR-QoL because the scores in all domains were better than the national norm (score, 50) in both groups. These findings suggest that when LSS symptoms occur, HR-QoL may worsen, and when LSS symptoms do not occur, HR-QoL may be maintained. These results were also almost the same as the subjects without either knee or hip OA and previous one-year follow-up (Appendix A) [11].

From a time course point of view, subjects who were LSS-positive at the initial analysis but LSS-negative at the six-year follow-up had a higher score of domains at the initial analysis than those who were LSS-positive in both the initial analysis and six-year follow-up, although only BP in all subjects had a statistical difference (Table 5). Moreover, subjects who were LSS-negative at the initial analysis but positive at the six-year follow-up had a lower score of all eight domains at the initial analysis than those who were LSS-negative in both the initial analysis and the six-year follow-up. Statistical differences were observed in five of the eight domains (except for SF, RE, and MH) (Table 5).

These findings were almost the same as the subjects without either knee or hip OA and similar to those at the previous one-year follow-up (Appendix A) [11].

### 3.5. Predictive Factors for LSS Symptom Presence and Risk Ratio of Leg Symptoms for LSS Symptom Development

Regarding the positive predictive factors for LSS symptom presence at the six-year follow-up, LSS symptom positive and lower norm-based RDQ score were detected in all subjects (Table 6). Male sex, LSS symptom positive, lower norm-based RDQ score, and knee OA in the initial analysis appeared to be predictive factors for LSS symptom presence during the six-year follow-up period in all subjects. Compared with the previous one-year follow-up study, LSS symptom positive and RDQ score still remained. Male sex and knee OA were newly identified [11]. Similarly in the subjects without either knee or hip OA, LSS symptom positive, lower norm-based RDQ score, and cardiovascular disease were detected (Appendix A).

Regarding the risk of leg symptoms resulting in LSS symptom development at the six-year follow-up, the subjects at the initial LSS-negative group who answered “yes” to one or more of Q1–6 in the LSS-SSHQ had 2.301 times higher risk among all subjects (Table 7). In the subjects without either knee or hip OA, the risk ratio was 17.30 (Appendix A).

Regarding the predictive factors of operation for LSS at the six-year follow-up, no statistically significant factor was identified (Table 8 and Appendix A).

## 4. Discussion

Currently, LSS is receiving a considerable amount of attention as a major health problem. Mainly, there are two suspected reasons for this attention; one is that the prevalence of LSS is increasing with age since most cases of LSS are caused by degenerative changes (acquired stenosis) [22,23,24,25,26,27], and the other is that LSS has a strongly negative influence on QoL compared with other comorbidities [1,22]. Although LSS symptoms typically have a negative impact on health, details of the natural course of LSS symptoms are still unclear in the community. 

### 4.1. Time Course of LSS Symptoms for the Six-Year Duration

At the one-year follow-up, the symptoms in 46% of subjects in the LSS-positive group improved. On the other hand, 10% of the subjects in the LSS-negative group at the initial analysis showed LSS symptoms at the one-year follow-up [11]. In the current study, which included subjects from the same cohort but over a six-year period, 56–65% of the subjects in the LSS-positive group at the initial analysis experienced improvement in their symptoms over the study period, whereas 12–15% of the subjects in the LSS-negative group at the initial analysis showed LSS symptoms at the six-year follow-up. There was no significant difference regarding the time course of LSS symptoms between all subjects and those without either knee or hip OA. When comparing the data obtained at the one- and six-year follow-ups, the time course of LSS symptoms was almost the same. In other words, the prevalence of subjects with LSS symptoms did not remarkably increase even after a period of six years. These results indicate that non-operative treatment should be considered the first choice for treatment of LSS.

### 4.2. Time Course of LSS Symptoms for the Six-Year Duration and QoL

LBP-related QoL (RDQ) and HR-QoL (eight domains of SF-36) showed partly different results in LSS symptoms changes at the six-year follow-up. For LBP-related QoL, when LSS symptoms improved, the average RDQ score also improved among all subjects and those without either knee or hip OA. On the other hand, regarding HR-QoL, even if LSS symptoms improved, most domains showed worsened outcomes. Furthermore, the average RDQ score and all eight domains were above the national average at the initial assessment and six-year follow-up if LSS symptoms were negative at the initial assessment and six-year follow-up. These results were similar to those at the previous one-year follow-up [11]. Overall, LBP-related QoL changed at the six-year follow-up with the changes in LSS symptoms; however, once LSS symptoms appeared, HR-QoL worsened regardless of whether said symptoms improved or persisted. These results suggest that LSS has a negative effect on HR-QoL, at least over a period of six years. In other words, even if patients have LSS symptoms but their QoL is not impaired, they may not have LSS symptoms at least six years later. Therefore, it can be said that family doctors should select conservative treatment for the first choice, if the patients’ QoL is not impaired.

### 4.3. Related and Predictive Factors for LSS symptom Presence or Development and Operation

In addition, similar to the one-year follow-up study, the current six-year follow-up study was still not fully successful in describing the possible related or predictive factors for LSS symptom presence [11]. Several comorbidities, such as diabetes mellitus and hypertension, may possibly be related or predictive factors for LSS symptom development as reported in previous studies [28,29]. In our previous study, RDQ score, knee OA, cerebrovascular disease, and smoking at the initial analysis were found to be related factors for LSS symptom development [11]. However, in the present study, norm-based RDQ score and knee OA remained as the related factors in the all subjects and norm-based RDQ score in the subjects without either knee or hip OA. Similarly, for the predictive factors of LSS symptom presence during the six-year follow-up period, male sex, LSS symptom positive, lower norm-based RDQ score, and knee OA at the initial analysis were detected in all subjects, and LSS symptom positive, lower norm-based RDQ score, and cardiovascular disease were detected in those without either knee or hip OA. When comparing these predictive factors with those identified at the one-year follow-up, LSS symptom positive and RDQ were identified in both studies including the subjects with knee and hip OA and in the subjects without either knee or hip OA of this six-year follow-up study, while other factors depended on the subject group and follow-up period [11]. These findings on predictive factors might mean that it is difficult to find specific and clear predictive factors for LSS symptoms, at least when using the current study’s protocol.

Among all subjects, those in the LSS symptom negative group with leg symptoms at the initial analysis were at more than double the risk of LSS symptom development at the six-year follow-up and were at more than triple the risk at the one-year follow-up compared to those without any leg symptoms [11]. On the other hand, in the subjects without either knee or hip OA, the risk ratio of the presence of leg symptoms was extremely high (17.30). Future studies should clarify the relationship between the presence of leg-symptoms at initial analysis and LSS symptom development at follow-up.

In the current study, the predictive factors for LSS surgery during the six-year follow-up period were not determined. Theoretically, these predictive factors should overlap with the predictive factors of the LSS symptom development and/or presence; however, it is well-known that the LSS symptom presence does not necessarily indicate a need for surgery. Furthermore, there are various factors involved in the process of determining whether to operate [30,31,32], including the degree of pain and symptoms, the individual’s way of thinking, and family issues. Therefore, the results regarding the predictive factors for surgery in the present study might be limited.

Overall, it goes without saying that if QoL declines due to persistent LSS symptoms, the family doctor should consult a specialist for further treatment such as physiotherapy, nerve blocking, surgery, and so on. It is also important to note that even if LSS cannot be clearly diagnosed but there are findings in the lower extremities that suggest neurological symptoms, it is likely that LSS symptoms will become apparent in the future.

### 4.4. Impact of Knee and Hip OA on LSS-SSHQ

In the present study, we used the LSS-SSHQ for many subjects to standardize the definition of LSS symptoms. Although the LSS-SSHQ was designed for use in epidemiologic studies and primary care settings, the results of the LSS-SSHQ may be influenced by the presence of knee or hip OA because LSS-SSHQ evaluates lower limb symptoms. In this study, we conducted the same analysis in both all subjects and those without either knee or hip OA. Although there was no overall significant difference in the results with or without either knee or hip OA, the risk ratio of developing LSS at the six-year follow-up was significantly different. These results indicate that LSS-SSHQ is acceptable for use in epidemiologic studies and primary care settings; however, it might be better to exclude knee or hip OA for better accuracy.

### 4.5. Limitations

There were several limitations to this study [1,11]. First, this research was conducted in a mountainous and rural area. Second, the subjects were volunteers. The research area and type of subjects may have resulted in selection bias. Third, the follow-up rate 50% might be acceptable considering the duration of follow-up period; however, it might not be sufficient. Fourth, LSS symptoms were defined by the questionnaire without imaging modalities such as MRI. Although a validation study was done in LSS-SSHQ (sensitivity; 84%, specificity; 78%), about 20% of the LSS-positives were suspected to be false positives [13]. Since the prevalence of LSS symptoms was about 20% in 2004 and 2010, false positives or false negatives could be misclassified, thus affecting the results. Similarly, the results of this study might change if imaging tests such as MRI are added in addition to the LSS-SSSHQ study of subjective symptoms. Fifth, this study investigated the presence or absence of LSS symptoms at six-year follow-up and does not indicate a full course of six-year LSS symptoms. Sixth, changes in comorbidities during the six-year follow-up period were not assessed. Seventh, LSS severity was not evaluated. Eighth, information on subjects who had surgery for their LSS was not detailed. The subjects were only asked if they had undergone surgery during the study period or not. Finally, there was no information on the duration of LSS symptoms and LSS treatment, such as medication, physical therapy and epidural injection. In spite of these limitations, the present study is still worth reporting because, to our knowledge, this is the biggest study for the time course of LSS symptoms in a community. We believe that the results of this study will help physicians decide on treatment strategies and explain when seeing LSS patients. However, because LSS is a chronic condition, the result of 6-year follow-up might still be preliminary. Further study is needed to investigate long-term follow-up LSS symptoms and its risk factors for the maintenance of health in the elderly in a community.

## 5. Conclusions

At the six-year follow-up, more than half of the subjects who were LSS-positive at the initial analysis became LSS negative, and 12–15% of those who were LSS-negative became LSS-positive. A relationship existed between the improvement and worsening of LSS symptoms and those of LBP-related QoL; however, such a clear relationship was not observed between LSS symptoms and almost all domains of SF-36 measured for HR-QoL. The subjects with LSS symptoms and low LBP-related QoL were more likely to have LSS symptoms at the six-year follow-up. However, no predictors were found that could lead to surgery for six-year duration.

## Figures and Tables

**Figure 1 medicina-57-01116-f001:**
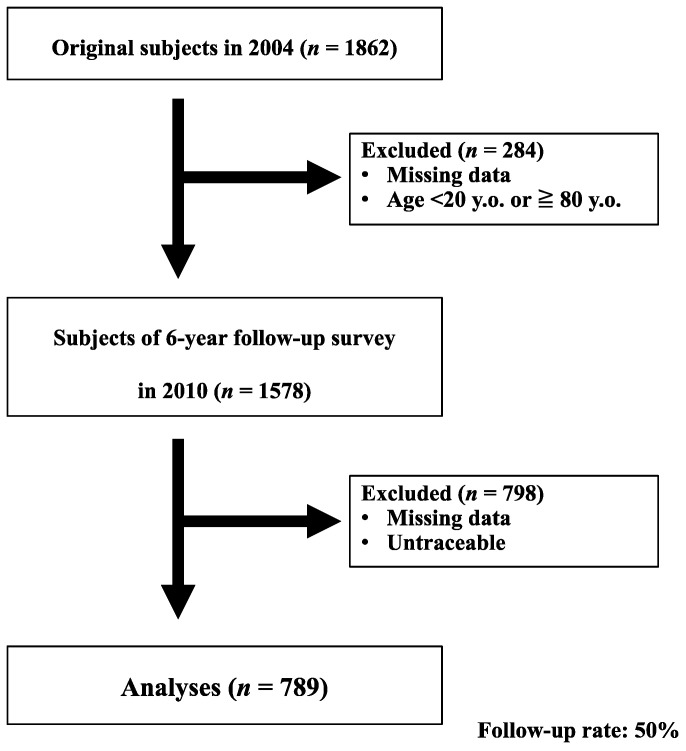
Participants.

**Figure 2 medicina-57-01116-f002:**
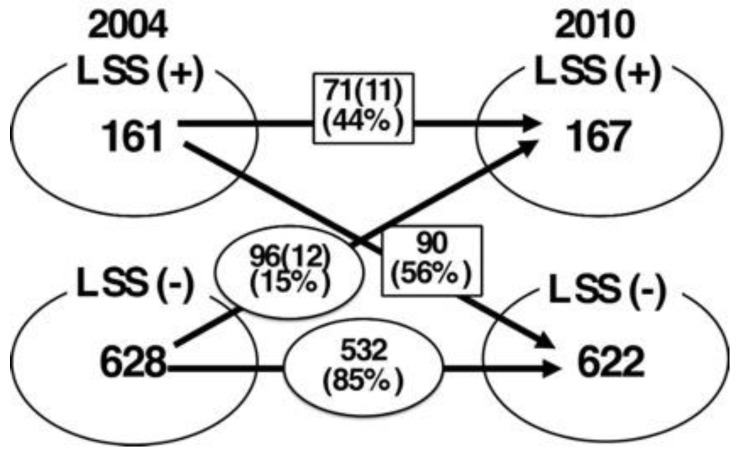
Time course of LSS symptom for six-year duration: All subjects.

**Table 1 medicina-57-01116-t001:** Proportion of participants in 2004 (initial assessment) and 2010 (six-year follow-up) by age group.

Age in 2004	Participants in 2004				Followed-Up Participants in 2010			
	All		Without Either Knee or Hip OA		All		Without Either Knee or Hip OA	
	Male	Total (*n* = 1578) (%)	Male	Total (*n* = 1079) (%)	Male	Total (*n* = 789) (%)	Male	Total (*n* = 513) (%)
40–49 y	60	135 (8.6)	57	124 (11.5)	19	37 (4.7)	18	35 (6.8)
50–59 y	89	253 (16.0)	75	201 (18.6)	30	108 (13.7)	25	83 (16.2)
60–69 y	178	537 (34.0)	149	375 (34.8)	102	329 (41.7)	81	218 (42.5)
70–79 y	235	653 (41.4)	180	379 (35.1)	111	315 (39.9)	89	177 (34.5)

Abbreviations: OA, osteoarthritis

**Table 2 medicina-57-01116-t002:** Demographic data in 2004 and 2010.

		2004		2010	
		All Subjects (*n* = 1578)	Subjects without Either Knee or Hip OA (*n* = 1079)	All Subjects (*n* = 789)	Subjects without Either Knee or Hip OA (*n* = 513)
Gender	Male:Female	562:1016	462:617	262:527	72:441
BMI	<18.5	72	54	36	23
	18.5–25.0	937	643	492	326
	25.1–30.0	413	272	199	124
	30<	33	24	15	11
Knee OA	Positive	450	-	257	-
Hip OA	Positive	73	-	34	-
Comorbidities	Respiratory	7	5	4	2
	Diabetes Mellitus	71	45	35	20
	Cardiovascular	136	80	67	34
	Cerebrovascular	21	12	7	2
	Hypertension	518	328	275	166
Smoking	Pack-Year ≥ 20	426	323	211	160
Depressivesymptoms	Severe	194	122	90	51
	Moderate	137	103	71	51
	Mild	173	109	94	60
	None	1065	738	528	346

Abbreviations: BMI, body mass index; OA, osteoarthritis.

**Table 3 medicina-57-01116-t003:** Change of LSS symptoms positive and LSS symptoms negative groups by age.

Age in 2004	Change of LSS Symptoms (*n* = 789)					
	LSS (+) in 2004 → 2010			LSS (−) in 2004 → 2010		
	(+) → (+)	(+) → (−)	*n*	(−) → (+)	(−) → (−)	*n*
40–49 y	2 (50.0)	2 (50.0)	4	2 (6.1)	31 (93.9)	33
50–59 y	9 (60.0)	6 (40.0)	15	9 (9.7)	84 (90.3)	93
60–69 y	23 (41.1)	33 (58.9)	56	33 (12.1)	240 (87.9)	273
70–79 y	37 (43.0)	49 (57.0)	86	52 (22.7)	177 (77.3)	229

Abbreviations: LSS, lumbar spinal stenosis.

**Table 4 medicina-57-01116-t004:** Change in RDQ scores of the subjects with/without LSS symptoms.

Change of LSS Symptoms (*n* = 766)	2004	2010	*p* Value
(+)→(+)	46.5 ± 8.4^(2)^	47.5 ± 8.6^(1)^	0.3251
(+)→(−)	50.1 ± 7.8^(2)^	52.1 ± 9.0^(1)^	0.1174
(−)→(+)	52.7 ± 8.4^(3)^	48.3 ± 10.5^(4)^	0.0014
(−)→(−)	55.7 ± 6.0^(3)^	56.6 ± 6.9^(4)^	<0.0001

Average ± SD ^(1)^; *p* < 0.05, ^(2)^; *p* < 0.01, ^(3)^; *p* < 0.001, ^(4)^; *p* < 0.0001. Abbreviations: LSS, lumbar spinal stenosis; RDQ, Roland-Morris Disability Questionnaire; SD, standard deviation.

**Table 5 medicina-57-01116-t005:** Changes in eight domains of the SF-36 of the subjects with/without LSS symptoms.

	Time Course of LSS (*n* = 766)	2004	2010	*p* Value
PF	(+)→(+)	48.1 ± 11.6	36.9 ± 14.2	0.0004
	(+)→(−)	50.1 ± 10.9	43.6 ± 12.5	0.0003
	(−)→(+)	48.5 ± 11.3^(1)^	42.1 ± 13.0^(5)^	0.0002
	(−)→(−)	52.9 ± 8.8^(1)^	51.5 ± 9.0^(5)^	0.0044
RP	(+)→(+)	44.3 ±9.9	37.0 ± 11.6^(1)^	0.0125
	(+)→(−)	47.2 ± 11.3	45.0 ± 11.4^(1)^	0.4468
	(−)→(+)	46.9 ± 10.8^(2)^	41.8 ± 10.7^(5)^	0.0137
	(−)→(−)	50.7 ± 9.4^(2)^	50.0 ± 9.4^(5)^	0.4047
BP	(+)→(+)	42.1 ± 7.7^(1)^	41.4 ± 6.7	0.5832
	(+)→(−)	46.4 ± 9.1^(1)^	44.5 ± 9.1	0.2359
	(−)→(+)	46.0 ± 11.9^(2)^	42.1 ± 8.5^(5)^	0.029
	(−)→(−)	51.7 ± 9.3^(2)^	51.3 ± 9.3^(5)^	0.5318
GH	(+)→(+)	46.0 ± 10.5	43.3 ± 8.2^(5)^	0.0795
	(+)→(−)	47.5 ± 10.1	47.0 ± 8.7^(5)^	0.9413
	(−)→(+)	48.2 ± 8.6^(3)^	46.7 ± 9.0^(5)^	0.2283
	(−)→(−)	52.0 ± 9.0^(3)^	51.8 ± 9.0^(5)^	0.9771
VT	(+)→(+)	47.0 ± 9.2	45.1 ± 9.0	0.2476
	(+)→(−)	50.4 ± 8.7	46.5 ± 9.6	0.014
	(−)→(+)	50.1 ± 10.1^(3)^	46.1 ± 9.7^(5)^	0.0254
	(−)→(−)	54.7 ± 8.2^(3)^	51.9 ± 8.5^(5)^	<0.0001
SF	(+)→(+)	46.1 ± 12.8	44.9 ± 11.6	0.4565
	(+)→(−)	49.9 ± 9.9	46.0 ± 10.8	0.049
	(−)→(+)	48.8 ± 11.6	44.8 ± 11.8^(3)^	0.006
	(−)→(−)	51.4 ± 9.1	50.4 ± 9.2^(3)^	0.1466
RE	(+)→(+)	45.3 ± 11.0	40.1 ± 12.0	0.0368
	(+)→(−)	48.2 ± 11.4	45.0 ± 11.6	0.2075
	(−)→(+)	45.6 ± 12.2	43.5 ± 11.0^(5)^	0.2882
	(−)→(−)	51.7 ± 8.5	50.5 ± 9.4^(5)^	0.6385
MH	(+)→(+)	46.4 ± 12.3	45.5 ± 11.9	0.5946
	(+)→(−)	47.3 ± 9.4	47.5 ± 9.2	0.8768
	(−)→(+)	47.7± 9.3	46.7 ± 10.5^(4)^	0.3866
	(−)→(−)	50.9 ± 9.1	52.0 ± 8.4^(4)^	0.0528

Average ± SD ^(1)^ *p* < 0.05, ^(2)^ *p* < 0.01, ^(3)^ *p* < 0.005 ^(4)^. Abbreviations: LSS, lumbar spinal stenosis; SF-36, the 36-Item Short Form Health Survey; SD, standard deviation; PF, physical functioning; RP, role-physical; BP, bodily pain; GH, general health perception; VT, vitality; SF, social functioning; RE, role-emotional; MH, mental health..

**Table 6 medicina-57-01116-t006:** Predictive factors for LSS symptom presence at the end of six-year follow-up period.

		OR	95%CI	*p* Value
Gender	Female	0.506	0.311–0.824	0.0061
Age	40–49 y	Reference		
	50–59 y	1.21	0.337–4.346	0.7703
	60–69 y	1.17	0.363–3.771	0.7929
	70–79 y	1.998	0.622–6.419	0.2451
BMI	<18.5	Reference		
	18.5–25.0	1.785	0.616–5.174	0.2859
	25.1–30.0	1.584	0.525–4.777	0.4145
	30<	0.606	0.056–6.583	0.6806
LSS symptoms	Positive	2.867	1.790–4.594	<0.0001
RDQ score (Norm-based)	<50	1.793	1.126–2.855	0.0139
Knee OA	Positive	2.03	1.287–3.203	0.0023
Hip OA	Positive	1.426	0.567–3.587	0.4509
Comorbidities	Respiratory	1.488	0.126–17.535	0.7024
	Diabetes Mellitus	1.972	0.844–4.611	0.117
	Cardiovascular	1.079	0.553–2.102	0.8242
	Cerebrovascular	1.72	0.340–8.716	0.5122
	Hypertension	1.51	0.984–2.319	0.0594
Smoking	Pack-Year ≥ 20	1.167	0.715–1.904	0.5368
Depressive symptoms	None	Reference		
	Mild	1.359	0.731–2.526	0.3321
	Moderate	1.168	0.553–2.468	0.6835
	Severe	1.389	0.720–2.678	0.3265

Abbreviations: LSS, lumbar spinal stenosis; BMI, body mass index; RDQ, Roland–Morris Disability Questionnaire; OA, osteoarthritis.

**Table 7 medicina-57-01116-t007:** Risk ratio of leg symptoms resulting in LSS development during the six-year follow-up period.

Initial Analysis		Development of LSS at Six-Year Follow-Up	
		LSS (+)	LSS (−)
**Leg symptoms in 2004**	(+)	53	166
	(−)	43	366

Abbreviations: LSS, lumbar spinal stenosis.

**Table 8 medicina-57-01116-t008:** Predictive factors of the operation for LSS during the six-year follow-up period.

		All Subjects		
		OR	95%CI	*p* Value
Gender	Female	0.555	0.192–1.604	0.277
Age	40–49 y	Reference		
	50–59 y	0.331	0.040–2.729	0.3043
	60–69 y	0.247	0.039–1.583	0.1402
	70–79 y	0.405	0.066–2.501	0.3305
BMI	<18.5	Reference		
	18.5–25.0	0.853	0.099–7.399	0.885
	25.1–30.0	1.034	0.112–9.525	0.9766
	30<	8.747 × 10^−6^	0	0.9982
LSS symptoms	Positive	2.906	0.953–8.861	0.0608
RDQ score (Norm-based)	<50	0.972	0.918–1.029	0.3348
Knee OA	Positive	1.056	0.350–3.186	0.9224
Hip OA	Positive	1.132	0.120–10.713	0.8201
Comorbidities	Respiratory	9.086 × 10^−8^	0	0.9989
	Diabetes Mellitus	5.257 × 10^−8^	0	0.9967
	Cardiovascular	0.683	0.138–3.383	0.6409
	Cerebrovascular	3.690 × 10^−8^	0	0.9985
	Hypertension	2	0.737–5.427	0.1736
Smoking	Pack-Year ≥ 20	0.817	0.256–2.610	0.7331
Depressive symptoms	None	Reference		
	Mild	1.572	0.410–6.028	0.5079
	Moderate	1.145	0.226–5.800	0.8702
	Severe	1.414	0.339–5.896	0.6341

## Data Availability

The data presented in this study could be available on request from the corresponding author. The data are not publicly available because the underlying data was obtained from the collaboration with the local government and contains sensitive information on individuals including gender, age and self-reported data, and sharing these data openly is prohibited by the local government and Fukushima Medical University Ethics Committee.

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
