# Peer review of "The Change of Lumbar Spinal Stenosis Symptoms over a Six-Year Period in Community-Dwelling People"

_medicina, 2021, doi:10.3390/medicina57101116_

Round 1

Reviewer 1 Report

This is a pretty an interesting study, however, some issues need to be addressed:

1 – in my opinion, the paper is too wordy. The analysis is complex and wide and not always expressed in a clear form. It is, in most of its part, challenging to follow. I will reword a few main concepts and present them more clearly.

2 – to me, it seems the proportion of patients LSS positive who improved over six years is high. Can the authors explain such findings? The lack of precise information about intervening therapy during the 6 years follow-up is a major limitation of this paper.

Also, why did such a low percentage of patients within LSS positive patients underwent surgery? Why physicians did not suggest LSS-positive patients seek medical advice.

3 – I commend the authors for having reported the numerous limitations of the paper; however, the lack of an imaging classification to correlate to clinical findings is a major drawback point and needs to be stated.

4 – why a multivariate analysis was not performed? The authors collected possible confounders and modifiers. Why not add them in an appropriate analysis of outcomes.

I would like to read a revised version of the paper.

Author Response

Reviewer 1

   Thank you very much for your interest in this research and for your constructive opinions.

1 – in my opinion, the paper is too wordy. The analysis is complex and wide and not always expressed in a clear form. It is, in most of its part, challenging to follow. I will reword a few main concepts and present them more clearly.

Response: As you pointed out, the text alone exceeds 4000 words. Honestly saying, this was the result of a previous revision of the one-year follow-up study following the reviewer's comments when submitting to a few medical journals. In this six-year follow-up, we tried to do the same analysis as the one-year follow-up. We agree with reviewer’s suggestion. For easy understanding of the readers, we focus on the time course of LSS symptom (Figure 2, Table 3) and QoL with the change of LSS symptoms (Table 4,5). For other analyses, we will try to reduce description and move Tables regarding analysis in the subjects without either knee or hip OA to Supplement. As a result, we succeeded in reducing about 5 pages.

2 – to me, it seems the proportion of patients LSS positive who improved over six years is high. Can the authors explain such findings? The lack of precise information about intervening therapy during the 6 years follow-up is a major limitation of this paper.

Response: Thank you very much for important comment. We think there might be two major reasons; One is LSS symptoms are relatively mild, not severe because participants are local residents and most of all might not be a patient in the hospital or clinics. This hypothesis seems to be supported by the fact that only 6.8% of the subjects in LSS symptom positive group had received operation in 6 years. Another is the study design-related factor. In this study, to standardize the LSS symptoms, we used LSS-SSHQ. LSS-SSHQ is a just a screening tool for LSS, therefore not all symptomatic LSS cases could be detected. In negative LSS-symptom group, the subjects who have some LSS symptoms but do not meet the LSS-SSHQ criteria are classified in the LSS negative group. So, we did the analysis such as Table 8 in the original version or Table 7 in the revision version.

3 – I commend the authors for having reported the numerous limitations of the paper; however, the lack of an imaging classification to correlate to clinical findings is a major drawback point and needs to be stated.

Response: Thank you very much for reviewers’ comments. As a researcher's sincere attitude, we think it is very important to clearly state the limits. We added the one sentence in Limitation; “Fourth, LSS symptoms were defined by the questionnaire without imaging modalities such as MRI. Although a validation study was done in LSS-SSHQ (sensitivity; 84%, specificity; 78%), about 20% of the LSS-positive were suspected to be false positives [13]. Since the prevalence of LSS symptoms was about 20% in 2004 and 2010, false positive or false negative could be misclassified, thus affecting the results. Similarly, the results of this study might change if imaging tests such as MRI are added in addition to the LSS-SSSHQ study of subjective symptoms.”(Line 484-486).

4 – why a multivariate analysis was not performed? The authors collected possible confounders and modifiers. Why not add them in an appropriate analysis of outcomes.

Response: Thank you very much for reviewers’ comments. We are very sorry and embarrassed that we used the term "multiple" where we should describe "multivariate" in a logistic regression analysis. Of course, all logistic regression analysis in this study was "multivariate" analysis. We correct it. (Line112)

Reviewer 2 Report

I had the pleasure to review the paper presented by Otani at al.

The aim of the paper is to clarify the longitudinal time course and associated factors of lumbar spinal stenosis symptoms over a period of six years in a community. 

The paper is relatively well written, material and methods are well ideated, conclusions are clear. However, the six-year time frame seems too short for such a chronic condition as lumbar spinal stenosis. 

Having said that, I would consider this more as a preliminary report than a final paper and I would be very interested in looking at the results at 15-20 years follow-up.

Author Response

Reviewer 2

Thank you very much for your interest in this research and for your constructive opinions.

1 –Having said that, I would consider this more as a preliminary report than a final paper and I would be very interested in looking at the results at 15-20 years follow-up.

Response: Thank you very much for your encouraging us. We will try to resurvey for 20-year follow-up. We also added the one sentence in Limitation; “However, because LSS is chronic condition, the result of 6-year follow-up might be still preliminary” (line 496-497).

Round 2

Reviewer 1 Report

I commend the authors for having replied to all of the previous comments.